# Modern contraceptive utilisation and its associated factors among reproductive age women in high fertility regions of Ethiopia: a multilevel analysis of Ethiopia Demographic and Health Survey

Tadele Biresaw Belachew [1], Wubshet Debebe Negash [1], Desalegn Anmut Bitew,[2] Desale Bihonegn Asmamaw [2]

[1]Department of Health Systems and Policy, University of Gondar, Gondar, Ethiopia
[2]Department of Reproductive Health, Institute of Public Health, University of Gondar, Gondar, Ethiopia

**Correspondence to**
Tadele Biresaw Belachew;
tadelebiresaw01@gmail.com

## ABSTRACT

**Objective** This study is aimed to assess the magnitude of modern contraceptives utilisation and associated factors among reproductive age women in high fertility regions of Ethiopia.

**Design** Cross-sectional study.

**Setting** High fertility regions of Ethiopian.

**Participants** A total weighted sample of 3822 married reproductive age women.

**Methods** In this study, data were obtained from the recent Ethiopian Demographic and Health Surveys. A total weighted sample of 3822 women of reproductive age was included. A multilevel mixed-effect binary logistic regression model was fitted to identify the significant associated factors of modern contraceptive utilisation. Statistical significance was determined using adjusted OR (AOR) with 95% CI.

**Results** The overall modern contraceptive utilisation was 29.75% (95% CI 28.2% to 31.2%). Among the factors associated with utilisation were women's age 25–34 years (AOR 1.3; 95% CI 1.01 to 1.66) and ≥35 (AOR 1.71; 95% CI 1.37 to 2.70), husband's occupation (AOR 1.49; 95% CI 1.03 to 1.99), number of alive children: 1–4 (AOR 2.20; 95% CI 1.47 to 3.30), 5–8 (AOR 1.74; 95% CI 1.09 to 2.77), husband's desired number of children (AOR 0.77; 95% CI 0.61 to 0.96), residency (AOR 2.37; 95% CI 1.20 to 4.67), community media exposure (AOR 1.77; 95% CI 1.01 to 3.08), region (AOR 0.13; 95% CI 0.03 to 0.52) and religion (AOR 0.49; 95% CI 0.36 to 0.66) were significantly associated with modern contraceptive utilisation.

**Conclusion** Modern contraceptives utilisation in high fertility regions of Ethiopia was low. Women age, husband occupation, number of living children, husband's desired number of children, residency, community media exposure, region and religion were significantly associated with modern contraceptive utilisation. Therefore, to improve the utilisation of modern contraceptives, public health policy makers should consider creating awareness through mass media, male involvement in family planning, as well as family planning programmes, should be encouraged in rural areas.

## STRENGTHS AND LIMITATIONS OF THIS STUDY

⇒ This study used most recent nationally representative data, which were collected validated and standard data collection tools.
⇒ This study employed multilevel analysis (advanced model) that accounts the correlated nature of Ethiopian Demographic and Health Surveys (EDHS) data in the determination of the estimate.
⇒ The cross-sectional nature of the study does not show the cause and effect relationship between the outcome and the independent factors.
⇒ Moreover, due to EDHS were secondary data, essential factors like attitude and knowledge about contraceptives, partner's perspectives on contraceptives, and fear of side effects were not available in the EDHS.

have children by teaching, advocating and offering birth control methods.[1] It also prevents pregnancy-related health risks in women, infant mortality, sexually transmitted infections and adolescent pregnancy, while slowing population growth.[2 3] Additionally, it is the best investment in the healthcare and well-being of women, children and communities.[2] Access to quality, affordable sexual and reproductive health services like contraceptives and information is crucial to ensure the rights and well-being of women and girls.[4]

The types of contraceptive methods used for family planning can be divided into modern contraception and traditional methods that limit or delay childbirth.[5] Modern contraceptive methods are more effective than traditional methods that include short-acting (injectable, pills, male condom, female condom, emergency contraceptives),[6–8] long-acting reversible (hormonal implants and intrauterine devices)[9–11] and permanent (tubal-ligation (female sterilisation) and

## BACKGROUND

Family planning services support people in making decisions about whether to

vasectomy (male sterilisation))[12–14] methods that exclude traditional methods which helps to limit number of children and prevent maternal and child morbidity and mortality.[15]

In spite of contraceptive value in improving maternal and child health, a large proportion of women do not use contraceptives with remarkable variations across geographical areas and within country. Though, in the past 20 years, the utilisation of modern contraceptives has increased slightly, from 54.4% in 20th to 57.4% in 21th in the world and 23.6% in 2008 to 28.5% in 2015 in Africa, respectively, as reported by the United Nations Population Fund.[16 17]

According to studies conducted in Asia, modern contraceptive use among married women in Nepal and Bangladesh was 36% and 81.27%, respectively.[18 19] Similarly, studies in Africa show the high levels of modern contraception utilisation in South Africa, Baringo Kenya and Nigeria at 39.9%, 32%, 31% and 92.7%, respectively.[17 20 21] Moreover, different studies conducted in Ethiopia also shows that the magnitude of modern contraceptive utilisation ranges from 9.1% to 46.9%.[22–25]

Based on studies conducted in developing countries, factors affecting the use of contraceptives included age, education level, parity, religion, knowledge about modern family planning methods and side effects, method approval by partners and employment status.[26–28] In addition, studies in Ethiopia found that age, residence, education level of the mother, couple discussions, perception of husband approval, discussions with health extension workers, psychological acceptance, desire for more children, monthly income and number of living children were all affected by modern family planning method use.[22 24 29 30]

In Ethiopia, a variety of strategies have been employed to increase the uptake of contraceptive methods over the last decade. Among the steps taken to increase contraceptive use was the implementation of health extension programmes to change attitudes and improve awareness among the community.[31 32] To remove health system barriers to the use of contraceptives, the health system expanded health centres and health posts and upgraded primary healthcare units.[31] In spite of these efforts made at the national level, the proportion of women who use contraceptive methods remains low.[33]

We; therefore, aimed to determine what factors are associated with use of modern contraceptives among women of reproductive age groups in high fertility areas of Ethiopia. It is hoped that the results of the study will help policy makers to make interventions that will help reduce maternal mortality and morbidity through speeding up the utilisation of modern contraceptives.

## METHODS
### Study design, period and setting
A community-based cross-sectional survey was conducted using secondary data in the 2016 Ethiopian Demographic and Health Surveys (EDHS),[34] which was conducted by the Central Statistical Agency in collaboration with the Federal Ministry of Health and the Ethiopian Public Health Institute, which was a national representative sample conducted from 18 January 2016 to 27 June 2016. There are nine regional states in Ethiopia (Tigray, Afar, Amhara, Oromia, Benishangul, Gambela, South Nation, Nationalities and Peoples' Region (SNNPR), Harari and Somali), and two administrative cities (Addis Ababa and Dire-Dawa), 611 districts and 15 000 Kebeles. Afar, Somali and Oromia regions were included in this study. These regions were selected because they are the high fertility rate regions in Ethiopia with fertility rates above 5.0, a value that is higher than the rate of 4.6 in Ethiopia and 2.47 worldwide.[35 36]

### Data sources
The data for these regions were gained from the official database of the Demographic and Health Survey (DHS) programme, www.measuredhs.com after authorisation was granted via online request by explaining the purpose of our study. We extracted dependent and independent variables from the woman record (IR file). DHS is a nationally representative household survey conducted by face-to-face interviews on a wide range of populations. Study participants were selected using a two-stage stratified sampling technique. Enumeration areas (EAs) were randomly selected in the first stage, while households were selected in the second stage.[37] We calculated the individual weight for women (v005) by multiplying the household weight (hv005) by the inverse of the individual response rate. Before analysis, individual sample weights are generated by dividing (v005) by one million to approximate the number of cases.[10 38 39] After exclusion of pregnant women and infecund during survey, a total weighted sample of 3822 married reproductive-age women were included from three regions in this study (figure 1).

### Variables and measurements
#### Dependent variable
The outcome variable was modern contraceptive utilisation. In the current study, a woman was considered as modern contraceptive method utiliser if she had been using at least one of the modern contraceptives (female sterilisation, male sterilisation, IUCD, injectable, implants, pills, male condom, female condom and emergency contraception) during EDHS data collection period. Whereas a woman was considered to be non-utiliser of the modern contraceptive method if she had been using traditional methods like rhythm method, lactation amenorrhoea method and withdrawal or if she had not been using any type of contraception during EDHS data collection period.[40]

#### Independent variables
Different independent variables were considered in this study to determine factors associated modern

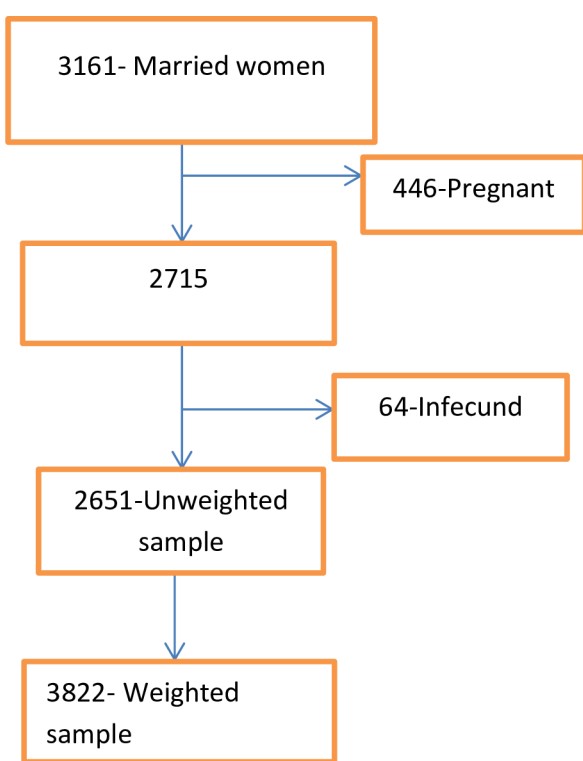

**Figure 1** Schematic illustration of women included in high fertility regions of Ethiopia.

**Table 1** List of variables for the assessment of modern contraceptive utilisation

| Variables | Description |
| --- | --- |
| Age of respondents | 15–24, 25–34, 35–49 |
| Educational status of respondents | No formal education, primary education, secondary and above |
| Husband education | No formal education, primary education, secondary and above |
| Occupation of respondents | Working, not working |
| Husband occupation | Working, not working |
| Wealth index | Poor, middle, rich |
| No of living children | None, 1–4, 5–8, ≥9 |
| History of abortion | No, Yes |
| Births in the last 3 years | No birth, one birth, Two and more birth |
| Visit of health facility in the last 12 months | Yes, no |
| Husband's desire for children | Both want the same, husband wants more, husband wants fewer, don't know |
| Media exposure | No, yes |
| Residence | Urban, rural |
| Community-level poverty | High, low |
| Community-level media exposure | Low, high |
| Community-level education | Low, high |
| Region | Somali, Afar, Oromia |
| Religion | Orthodox Christian, Muslim, Protestant, others |
| Distance to the health facilities | Big problem, not big problem |

contraceptive utilisation. Community-level variables, residences, region, distance to the health facilities and religion were directly accessed from EDHS data sets. However, community-level poverty, community-level education and community-level media exposure were constructed by aggregating individual-level characteristics at the cluster level.[41] They were categorised as high or low based on the distribution of the proportion values generated for each community after checking the distribution by using the histogram. The aggregate variable was not normally distributed and the median value was used as a cut-off point for the categorisation (table 1).[41 42]

### Data analysis
For data analysis Stata V.16 software was used. To ensure the representativeness of the EDHS sample and obtain reliable estimations and standard errors, data were weighted (v005/1000000) throughout analysis.

Four models fitted: the null model with no explanatory variables, model I with individual factors, model II with community factors, and model III with both individual and community factors. To compare and assess the fitness of nested models, we used the intraclass correlation coefficient (ICC), the median OR and deviation (−2LLR). Model III was the best-fitting model due to its low deviance. In multivariable analysis, variables with a p value less than 0.2 in bivariable analysis were used. Finally, in the multivariable analysis, adjusted ORs (AOR) with 95% CIs and p values less than 0.05 were used to identify factors of modern contraceptive utilisation.

### Patient and public involvement statement
No patient or the public was directly involved in developing the research questions, the design, protocol, data collection tools, results and dissemination plan of the study.

### RESULTS
#### Background characteristics of the study participants
A total weighted sample of 3822 individual women participated in this study.

Out of the total respondents, 2489 (65.12 %) women were not attended formal education, 2648 (69.28%) of respondents had no occupation and 2761 (72.24%) of the respondents had no media exposure about family planning. Among the participants, 2071 (54.17%) had 1–4 number of alive children. With regard to their economic status, 1626 (42.55%) women were from the poor wealth quintiles (table 2).

**Table 2** Individual characteristics of respondents in high fertility regions of Ethiopia (n=3822)

| Variables | Categories | Frequency | % |
|---|---|---|---|
| Age of respondents | 15–24 | 891 | 23.30 |
| | 25–34 | 1689 | 44.19 |
| | 35–49 | 1242 | 32.50 |
| Educational status of respondents | No formal education | 2489 | 65.12 |
| | Primary education | 1204 | 31.49 |
| | Secondary and higher | 130 | 3.39 |
| Husband education | No formal education | 1795 | 46.97 |
| | Primary education | 1503 | 39.33 |
| | Secondary and higher | 524 | 13.70 |
| Occupation of respondents | Not working | 2648 | 69.28 |
| | Working | 1174 | 30.72 |
| Husband occupation | Not working | 569 | 14.89 |
| | Working | 3253 | 85.11 |
| Wealth index | Poor | 1626 | 42.55 |
| | Middle | 749 | 19.59 |
| | Rich | 1447 | 37.86 |
| Media exposure | No | 2761 | 72.24 |
| | Yes | 1061 | 27.76 |
| No of living children | None | 251 | 6.56 |
| | 1–4 | 2071 | 54.17 |
| | 5–8 | 1339 | 35.02 |
| | ≥9 | 162 | 4.24 |
| History of abortion | No | 216 | 55.18 |
| | Yes | 175 | 44.82 |
| Births in the last 3 years | No birth | 1502 | 39.28 |
| | One birth | 1885 | 49.31 |
| | Two and more births | 436 | 11.41 |
| Visit of health facility in the last 12 months | No | 2187 | 57.22 |
| | Yes | 1635 | 42.78 |
| Husband's desire for children | Both want the same | 1298 | 34.03 |
| | Husband wants more | 1162 | 30.45 |
| | Husband wants fewer | 239 | 6.27 |
| | Don't know | 1116 | 29.26 |

**Table 3** Community-level characteristics of respondents in high fertility regions of Ethiopia (n=3822)

| Variables | Categories | Frequency | % |
|---|---|---|---|
| Residence | Rural | 3390 | 88.69 |
| | Urban | 432 | 11.31 |
| Community-level poverty | High | 750 | 19.62 |
| | Low | 3072 | 80.38 |
| Community media exposure | Low | 1405 | 36.76 |
| | High | 2417 | 63.24 |
| Community-level education | Low | 1214 | 31.77 |
| | High | 2608 | 68.23 |
| Region | Afar | 82 | 2.15 |
| | Oromo | 3485 | 91.18 |
| | Somali | 255 | 6.67 |
| Religion | Orthodox Christian | 756 | 19.77 |
| | Protestant | 693 | 18.13 |
| | Muslim | 2240 | 58.61 |
| | Others | 133 | 3.48 |
| Distance to health facilities | Big problem | 2235 | 58.48 |
| | Not big problem | 1587 | 41.52 |

respondents 3485 (91.18%) were from Oromia region. About 3072 (80.38 %) of the respondents were from communities with low proportion of poverty level. Almost two-thirds (63.24%) of women had media exposure and 2235 (58.48%) of participant's perceived as big problem to access health facilities (table 3).

## Modern contraceptive utilisation

Overall, the magnitude of modern contraceptive utilisation in high fertility regions of Ethiopia was 29.75% (95% CI 28.2% to 31.2%), with Oromia recording the highest magnitude of 32.18% (figure 2).

## Multilevel analysis of factors

In terms of the individual-level factors, the study showed that reproductive age women aged 25–34 and 35–49 were more likely to use modern contraceptive (AOR 1.3; 95% CI 1.01 to 1.66), (AOR 1.71; 95% CI 1.37 to 2.70) than compared with those aged 15–24 women, respectively. Women who had a husband with occupation during the time of survey had higher odds (AOR 1.49; 95% CI 1.03 to 1.99) to use modern contraceptive compared with who did not work. Women who had 1–4 number of living children were 2.2 times modern contraceptive use (AOR 2.20; 95% CI 1.47 to 3.30) than who had no children and reproductive age women who had 5–8 living children were more likely to use modern contraceptive (AOR 1.74; 95% CI 1.09 to 2.77) compared with women who had no living children. Women who had a partner whose desire

## Community-level factors

Of the respondents 3390 (88.69%) were rural dwellers. Nearly 60% respondents were Muslims. Among the

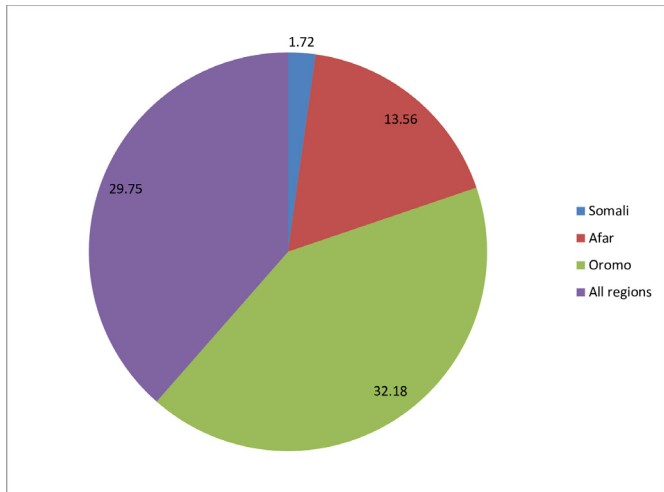

**Figure 2** Magnitude of modern contraceptive utilisation in high fertility regions of Ethiopia.

for children to have more were 23% less likely to use modern contraceptive (AOR 0.77; 95% CI 0.61 to 0 .96) compared with both want the same to have children and also women who had a Partner whose desire for children unknown were 29% less likely to use modern contraceptive (AOR 0.71; 95% CI 0.56 to 0.89) than both want the same to have children.

Regarding community-level factors, the odds of modern contraceptive utilisation were high among reproductive age women who had community media exposure (AOR 1.77; 95% CI 1.01 to 3.08) compared with their counterparts. Women who were residing in urban area were 2.37 times contraceptive use than living in rural area (AOR 2.37; 95% CI 1.20 to 4.67). In addition, women in Somali region were 87% less likely (AOR 0.13; 95% CI 0.03 to 0.52) to use modern contraceptive compared with women in Oromia region. Moreover, women who follow Muslim in religion were 51% less likely to use modern contraceptive (AOR 0.49; 95% CI 0.36 to 0.66) than Orthodox Christian followers (table 4).

### Measures of variation

According to the result in the null model, the ICC was 41.1% of the variations of modern contraceptive utilisation among study subjects were attributed by difference at the cluster level, but the rest 58.9% were attributed to individual women factors. The final model indicates that the proportional change in variance value of 0.606 indicates that both individual-level and community-level factors were responsible for about 60.6% of the variation in modern contraceptive utilisation between study subjects. When comparing models/fitness, deviance was used, this model was the best-fitting model, having the lowest deviance (3551) (table 4).

### DISCUSSION

The aim of this study was to investigate the magnitude and associated factors of modern contraceptive utilisation among reproductive age women in high fertility regions of Ethiopia. Slightly above one-fourth 29.75% (95% CI 28.2% to 31.2%) of participants were currently using modern contraceptive. This study is in line with study conducted in Dembia District.[43] The finding was lower than that reported from 2016 EDHS 35%,[44] study done in Adama Town 47% and Ethiopia 51.6%.[45 46] However, this finding is higher than the previous studies conducted in Bale,[22] Gumuz,[47] Rural women in Ethiopia.[48] The difference might be due to the different design and sample size.

In this study, the usage of contraceptives was strongly related with women of reproductive age, specifically those age 25–34 and >35 were more likely to use contraceptives than those age 15–24. The findings were similar to those obtained by Nepal.[27] This may be because younger women (15–24 years) perceive pregnancy risks are low.

The use of contraception by a wife is determined by her husband, and the wife is more likely to use contraception if her husband approves.[49] In this study, women have a higher likelihood of using contraception if their husband holds a professional, technical or managerial position. This husband's occupation may reflect husband's educational status and contributed for contraceptive use by women as indicated.[50] There is a higher possibility that these women can reside in urban regions.

The use of contraceptive methods was independently associated with the number of children. In comparison to a woman who had no children, women who had 1–4 children and women who had 5–8 children were nearly 2.2 and nearly 2 times more likely, respectively, to have used a modern form of contraception. Additionally, prior studies have shown that using modern contraceptives is more likely when there are a lot of living children.[34 51–55] This might be because women without children might need to have children in order to have the ideal number of children.[56]

The current study also showed that the husband's desire to have more children was inversely related to modern contraceptive use. Women with husbands who want many children were 23% less likely to use modern contraceptives than women who want the same number of children with their husbands. This is congruent with the results of secondary data analyses of the DHS of Bangladesh, Burkina Faso and Mali, which showed that a husband's desire to have children influenced a woman's use of modern contraception when she was of reproductive age.[57 58]

In regard to community-level factors, according to this study, residence was independently associated with modern contraceptive use. Urban women of reproductive age are about three times more likely than rural women to use contemporary contraceptives. This finding is in line with secondary data analysis of Indian, Afghan, Nigerian and Bangladeshi DHS, which showed that urban resident women were more likely than rural resident women to use contemporary contraceptives.[48 59–61] This could be caused by a variety of factors. Urban women tend to be

**Table 4** Multivariable analyses for factors affecting modern contraceptive utilisation (n=3822)

| Variables | Model 0 | Model 1 AOR (95% CI) | Model 2 AOR (95% CI) | Model 3 AOR (95% CI) |
|---|---|---|---|---|
| Individual-level characteristics | | | | |
| Age of respondents | | | | |
| 15–24 | | 1 | | 1 |
| 25–34 | | 1.39 (1.08 to 1.77) | | **1.29 (1.01 to 1.66)\*** |
| 35–49 | | 0.58 (0.43 to 0 .79) | | **1.71 (1.37 to 2.70)\*** |
| Educational status of the respondents | | | | |
| No formal education | | 1 | | 1 |
| Primary education | | 1.15 (0.93 to 1.44) | | 1.06 (0.85 to 1.33) |
| Secondary and higher | | 1.33 (0.78 to 2.30) | | 1.02 (.59 to 1.79) |
| Husband education | | | | |
| No formal education | | 1 | | 1 |
| Primary education | | 1.14 (0.93 to 1.5) | | 1.05 (0.85 to 1.29) |
| Secondary and higher | | 1.07 (0.76 to 1.47) | | 0.96 (0.68 to 1.34) |
| Husband occupation | | | | |
| Not working | | 1 | | 1 |
| Working | | 1.55 (1.16 to 2.06) | | **1.49 (1.03 to 1.99)\*** |
| Wealth index | | | | |
| Poor | | 1 | | 1 |
| Middle | | 1.24 (0.96 to 1.59) | | 1.02 (0.79 to 1.31) |
| Rich | | 1.53 (1.19 to 1.98) | | 1.19 (0.92 to 1.54) |
| Media exposure | | | | |
| No | | 1 | | 1 |
| Yes | | 1.24 (0.99 to 1.54) | | 1.15 (0.92 to 1.43) |
| No of living children | | | | |
| None | | 1 | | 1 |
| 1–4 | | 2.18 (1.46 to 3.25) | | **2.20 (1.47 to 3.30)\*** |
| 5–8 | | 1.70 (1.08 to 2.69) | | **1.74 (1.09 to 2.77)\*** |
| ≥9 | | 1.10 (0.51 to 2.39) | | 1.21 (0.55 to 2.65) |
| Visit of health facility in the last 12 months | | | | |
| No | | 1 | | 1 |
| Yes | | 1.32 (1.10 to 1.58) | | 1.29 (0.74 to 1.54) |
| Partner's desire for children | | | | |
| Both want the same | | 1 | | |
| Husband wants more | | 0.75 (0.61 to 0.94) | | **0.77 (0.61 to 0 .96)\*** |
| Husband wants fewer | | 0.84 (0.58 to 1.22) | | 0.82 (0.56 to 1.18) |
| Don't know | | 0.72 (0.57 to 0 .90) | | 0.71 (0.56 to 1.89) |
| Community-level variables | | | | |
| Residency | | | | |
| Rural | | | 1 | 1 |
| Urban | | | 2.86 (1.48 to 5.55) | **2.37 (1.20 to 4.67)\*** |
| Community-level media exposure | | | | |
| Low | | | 1 | 1 |
| High | | | 1.92 (1.09 to 3.39) | **1.77 (1.01 to 3.08)\*** |
| Community-level poverty | | | | |
| High | | | 0.32 (0.15 to 0.69 | 0.33 (0.15 to 1.70) |

Continued

**Table 4** Continued

| Variables | Model 0 | Model 1 AOR (95% CI) | Model 2 AOR (95% CI) | Model 3 AOR (95% CI) |
|---|---|---|---|---|
| Low | | | 1 | **1** |
| Community-level education | | | | |
| Low | | | 1 | 1 |
| High | | | 0.81 (0.45 to 1.43) | 0.75 (0.43 to 1.31) |
| Region | | | | |
| Oromo | | | 1 | 1 |
| Afar | | | 2.25 (0.87 to 5.82) | 2.19 (0.84 to 5.7) |
| Somali | | | 0.11 (0.03 to 0.44) | **0.13 (0.033 to 0.52)*** |
| Religion | | | | |
| Orthodox Christian | | | 1 | 1 |
| Protestant | | | 1.11 (0.83 to 1.50) | 1.08 (0.79 to 1.47) |
| Muslim | | | 0.52 (0.38 to 0 .71) | **0.49 (0.36 to 0.66)*** |
| Others | | | 0.61 (0.33 to 1.15) | 0.61 (0.32 to 1.16) |
| Random effect results | | | | |
| Variance (%) | 43.7 | 32.9 | 17.8 | 17.2 |
| ICC (%) | 41.1 | 33.3 | 20.6 | 19.5 |
| MOR | 17.1 | 14.8 | 10.9 | 10.7 |
| PCV | Ref | 24.7 | 59.3 | 60.6 |
| Model comparison | | | | |
| Deviance(−2LLR) | 3893 | 3747 | 3735 | 3551 |

*p<0.05.
AOR, adjusted OR; ICC, intraclass correlation coefficient; MOR, median OR; PCV, proportional change in variance.

more educated, earn more money, have better access to health facilities and have better media access than rural women, all of which lead to higher modern contraceptive utilisation rates. Women in rural areas also require more children to help them with field work, which negatively impacts their use of modern contraceptives.[40 62–66]

Women who were exposed to the community media had a higher likelihood of using modern methods of contraception than women who were not. The finding is in line with the results.[34 48 52 54 67] This is due to the possibility that exposure to mass media could play a significant role in raising awareness and inspiring women to take contemporary contraceptives.

In addition woman living in Somali region of Ethiopia was 87% less likely to use modern contraceptive compared with women in Oromia region. The reason could be different access to health information and different availability of maternal health services like family planning.[68]

Moreover, women who were Muslim religious followers were less likely to use modern contraceptive than Orthodox Christian religious followers. It is possible that the low rate of family planning methods, such as the usage of contemporary contraceptives, among Muslims and Protestants in Ethiopia is the cause of this.[69] In addition, there is evidence that Muslims are less likely to approve of the use of contraceptives and have a negative attitude about family planning.[70]

### Strengths and limitations

This study used most recent nationally representative data, which were collected with validated and standardised data collection tools. This also employed multilevel analysis (advanced model) that accounts the correlated nature of EDHS data in the determination of the estimate. Despite the above advantages, the cross sectional nature of the study does not show the cause and effect relationship between the outcome and the independent factors. Moreover, due to EDHS were secondary data, essential factors like attitude and knowledge about contraceptives, partner's perspectives on contraceptives, and fear of side effects were not available in the EDHS.

### CONCLUSION

Modern contraceptives utilisation in high fertility regions of Ethiopia was low. Women age, husband occupation, number of living children, husband's desired number of children, residency, community media exposure, region and religion were significantly associated with modern contraceptive utilisation. Therefore, to improve the utilisation of modern contraceptives, public health policy makers should consider creating awareness through mass media, male involvement in family planning, as well as family planning programmes, should be encouraged in rural areas.

**Acknowledgements** We are grateful to the EDHS programs, for the permission to use all the relevant EDHS data for this study.

**Contributors** TBB conceived the idea, extract the data, data analysis and draft the manuscript. WDN, DAB and DBA participate in the data analysis, interpretation and revising of the manuscript. All authors have read and approved the final manuscript. Guarantor, TBB.

**Funding** The authors have not declared a specific grant for this research from any funding agency in the public, commercial or not-for-profit sectors.

**Competing interests** None declared.

**Patient and public involvement** Patients and/or the public were not involved in the design, or conduct, or reporting, or dissemination plans of this research.

**Patient consent for publication** Not applicable.

**Ethics approval** Not applicable/no human participants included. Consent to participants is not applicable since the data are secondary and are available in the public domain. All the methods were conducted according to the Declaration of Helsinki. More details regarding DHS data and ethical standards are available online at (http://www.dhsprogram.com).

**Provenance and peer review** Not commissioned; externally peer reviewed.

**Data availability statement** Data are available on reasonable request. Data can be obtained on reasonable request. The data used in this study will be made available to the corresponding author on reasonable request.

**ORCID iDs**
Tadele Biresaw Belachew http://orcid.org/0000-0003-1738-6029
Wubshet Debebe Negash http://orcid.org/0000-0001-9720-7558
Desale Bihonegn Asmamaw http://orcid.org/0000-0001-7302-2575

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
