## [Reviewer comments · BMJ Open]

ARTICLE DETAILS

TITLE (PROVISIONAL)	Modern contraceptive utilization and its associated factors among reproductive age women in high fertility regions of Ethiopia: A multilevel analysis of Ethiopia Demographic and Health Survey
AUTHORS	Belachew, Tadele; Negash, Wubshet Debebe; Bitew, Desalegn Anmut; Asmamaw, Desale

VERSION 1 – REVIEW

REVIEWER	Myra Taylor Univrsity of KwaZulu-Natal , Public Health Medicine
REVIEW RETURNED	09-Sep-2022

GENERAL COMMENTS	BMJ OPEN REVIEW Modern contraceptive utilization and its associated factors among reproductive age women in high fertility regions of Ethiopia: A multilevel analysis of Ethiopia Demographic and Health Survey Thank you for the invitation to review this manuscript. The authors have chosen an important topic and have answered their objective: ‘to assess the magnitude of modern contraceptives utilization 23 and associated factors among reproductive age women in high fertility regions of Ethiopia’. Abstract: The abstract provides an useful summary of the study. Introduction. The authors have provided the background to this study. This includes four reported studies that provided the following information: ‘ In addition, studies in Ethiopia found that age, residence, education level of the mother, couple discussions, perception of husband approval, discussions with health extension workers, psychological acceptance, desire for more children, monthly income, and number of living children were all affected by modern family planning method use.13,15,20,21 Methods. The authors used secondary data, namely the DHS survey of 2016. The respondents to the survey are carefully sampled and the authors have used a large study sample (a weighted sample of nearly 4000 women) have explained which data were used from the survey Results. The authors have considered individual level associated factors and also analysed those at a community level, and provided adjusted odds ratios with 95% confidence intervals. Discussion. The authors have carefully discussed the presented results and compared their findings with other countries in Africa.
---

	Conclusion. The authors conclude: ‘Modern contraceptives utilization in high fertility regions of Ethiopia was low. Women’s age, husband occupation, number of living children, husband’s desired number of children, residency, community media exposure, region, and religion were significantly associated with modern contraceptive utilization. Therefore, to improve the utilization of modern contraceptives, public health policymakers should consider both individual and community level factors when designing family planning programs emphasis should be given to high risk women.’ REVIEWER: The conclusion is valid and the authors have done a really good job in performing the analysis and writing this paper. But I have a major problem with the concept. The authors’ findings are not very different from the quoted literature and the questions asked in the DHS survey are too broad to understand the issues. It is now 2022 and Africa knows all the above and this study does not really present anything new. Since Africa knows that women often face pressure when they wish to use contraception there are many practical questions such as access to a clinic; distance from the clinic; attitudes of the health service providers, stock-outs in contraceptive pills/injections etc. If we want women to be able to use modern contraception we must ensure that they are able to access such at a local level and discretely if need be. That requires asking different questions and working with the health authorities to implement what is required. The authors are well situated to do this and my challenge to them is to get out the office, move into the community and work with the healthcare providers to assist women in a patriarchal society such as Ethiopia to obtain the contraception that they need and deserve. I support the publication of this paper because it reinforces the importance of a different and more effective approach. ENGLISH EDITING: If the article was in Word it would be much easier to edit. It is easily readable but does need English home language editing.
--	---

REVIEWER	Rahma Mohamed Al Kindi Sultan Qaboos University
REVIEW RETURNED	20-Dec-2022

GENERAL COMMENTS	The research topic is very important although it is somewhat repetitive. The authors have adhered to the structure of writing the manuscript, but the language needs extensive review and editing.  1. The conclusion in the abstract needs to be improved and rewritten 2. In general, need to be cautious in using abbreviations e.g. STIs, EDHS as they may not be clear for all readers from different parts of the World. 3. In the background, some of the paragraphs and phrases are superficial and needs to be clarified and discussed in more depth e.g paragraph 2 4. In the method section, there is confusion on how the data were selected and what is the sample size and the weighted sample. 5. The definition of modern contraception in the introduction contradicts with the outcome variables where they included standard days method as one of the modern methods. 6. Patient and public involvement, this section is not clear 7. Ethical considerations, even though no consent is required but ethical approval to obtain data collection and analysis is still required 8. The discussion section, is well organized with good flow.
--

	However, it is very superficial and needs deeper and broader critique
--	---

VERSION 1 – AUTHOR RESPONSE

	Reviewer 1	Dear reviewer, thank you in advance for your consideration of our paper for publication. We authors revised all the reviewer comments. Kindly see the clean version of the manuscript.
1	A limitation is the lack of questions about access to health services which is a critical issue.	Dear reviewer thank you in advance for your important comments and susgggestion. In fact, you are correct. Access to health-care organizations is a critical issue for modern contraceptive use that should be investigated. However, because the data is secondary, there are no additional variables concerning access to health services. Having this we authors include previously missed variable as your recommendation, which is “distance to health facilities” classified as “a big problem” or “not a big problem” in EDHS data, though, it does not associate with modern contraceptive utilization. According to the descriptive analysis 58.48% of participants perceived access to health care facilities as a major issue (big Problem). Kindly see the clean version manuscript (Lines 142-143 and through out the result section).
	Reviewer 2	Dear reviewer, thank you in advance for your consideration of our paper for publication. We authors

		revised all the reviewer comments. Kindly see the clean version of the manuscript.
1	The conclusion in the abstract needs to be improved and rewritten	Dear reviewer thank you in advance for your important comments and suggestion. As a result of your recommendations and suggestions, we authors revised and wrote the abstract's conclusion. Kindly see the clean version manuscript (Lines 43-46).
2	In general, need to be cautious in using abbreviations e.g. STIs, EDHS as they may not be clear for all readers from different parts of the World.	My sincere thanks go out to inadvance for his comments and suggestions. We have revised and taken the correction throughout the manuscript. Kindly see the clean version of the manuscript
3	In the background, some of the paragraphs and phrases are superficial and needs to be clarified and discussed in more depth e.g paragraph 2	Thank you for your constructive comments and suggestions We authors revised the background section as your recommendation. Kindly see the clean version of the manuscript (Lines 67-73)
4	In the method section, there is confusion on how the data were selected and what is the sample size and the weighted sample.	Your constructive feedback and ideas are greatly appreciated. We authors revised the methods section according to your comments. Kindly see the clean version of the manuscript (Lines 124-129 and figure 1)

5	The definition of modern contraception in the introduction contradicts with the outcome variables where they included standard day's method as one of the modern methods.	Your constructive comments and suggestions are greatly appreciated. Sorry for typing error. We have omitted the term standard day's method which contradicts to bac ground section. Kindly see the clean version of the manuscript (Line 134-135).
6	Patient and public involvement, this section is not clear	Thank you for your constructive comments and suggestions. We authors revised according to your comments. kindly see the clean version of the manuscript (Lines 170-171)
7	Ethical considerations, even though no consent is required but ethical approval to obtain data collection and analysis is still required	Thank you for your important suggestion. We authors attempted to revise the ethics section and made changes based on the comments. Kindly see the clean version of the manuscript (Lines 164-168)
8	The discussion section is well organized with good flow. However, it is very superficial and needs deeper and broader critique	Thank you we have taken the correction Kindly see the clean version of the manuscript